# Lung Function Decline in Adult Asthmatics—A 10-Year Follow-Up Retrospective and Prospective Study

**DOI:** 10.3390/diagnostics11091637

**Published:** 2021-09-07

**Authors:** Salvatore Bucchieri, Pietro Alfano, Palma Audino, Fabio Cibella, Giovanni Fazio, Salvatore Marcantonio, Giuseppina Cuttitta

**Affiliations:** 1Institute for Biomedical Research and Innovation, National Research Council of Italy, Via U.La Malfa, 153, 90146 Palermo, Italy; salvatore.bucchieri@irib.cnr.it (S.B.); pietro.alfano@irib.cnr.it (P.A.); palma.audino@hotmail.it (P.A.); fabio.cibella@irib.cnr.it (F.C.); giuseppina.cuttitta@irib.cnr.it (G.C.); 2Triolo Zanca Clinic, Piazza Fonderia, 23, 90133 Palermo, Italy; 3Quality, Planning and Strategic Support Area, University of Palermo, Piazza Marina, 61, 90133 Palermo, Italy; salvatore.marcantonio@unipa.it

**Keywords:** asthma, FEV_1_ decline, exacerbation, reversibility, inhaled steroids

## Abstract

Asthma may have an impact on lung function decline but conflicting results are reported in forced expiratory volume in one second (FEV_1_) decline. We aimed to describe the changes in FEV_1_ in lifelong non-smoking adult asthmatic outpatients during a 10-year follow-up comparing years 1–5 (1st period) with years 6–10 (2nd period) to assess factors affecting these changes. A total of 100 outpatients performed spirometry every 3 months during a 10-year survey. FEV_1_/Ht^3^ slope values of the 2nd period reduced significantly respect to the 1st period (*p* < 0.0001). FEV_1_ slopes of years 1–5 and 6–10 were inversely associated with FEV_1_ at enrolment (*p* = 0.02, *p* = 0.01, respectively). Reversibility and variability FEV_1_ showed a significant effect on the 1st period slopes (*p* = 0.01 and *p* < 0.04, respectively). Frequent exacerbators in the 1st year had steeper FEV_1_/Ht^3^ slopes in the 1st period (*p* = 0.01). The number of subjects using higher doses of ICS was significantly lower at the 10th years respect to the 5th and the 1st year (*p* < 0.001, *p* = 0.003, respectively). This study shows that FEV_1_ decline in treated adult asthmatics non-smokers, over 10-year follow-up, is not constant. In particular, it slows down over time, and is influenced by FEV_1_ at enrolment, reversibility, variability FEV_1_ and exacerbation score in the 1st year.

## 1. Introduction

Conflicting results exist regarding lung function decline rates in asthmatics: in forced expiratory volume in one second (FEV_1_), rates range from almost normal [1,2] to those expected in chronic obstructive pulmonary disease (COPD) [3,4]. Asthma may have a significant impact on lung function decline [5,6] and recent studies suggest that long-term treatment with inhaled steroids may decrease FEV_1_ decline [7,8]. However, if little is known about other factors associated with FEV_1_ decline in adult asthmatics, the relationship of the characteristics of such patients and biomarkers of progression for airflow limitation, a functional consequence of airway remodelling, is considered important in the management of asthma [9]. We previously reported that longer disease duration may decrease the decline rate in asthmatics during a five-year period [10]: this interaction may add to the effects of asthma treatment.

We studied a sample of 100 non-smoker asthmatic outpatients followed-up by spirometry every 3 months during a 10-year survey. The purpose of this study was to evaluated the longitudinal FEV_1_ changes during this period, comparing years 1–5 (1st period) with years 6–10 (2nd period) also determining possible factors affecting functional decline.

## 2. Methods

A database was established in 2001. In this database were recorded the longitudinal data over the present: patients’ demographic characteristics, asthma symptoms based on Global Initiative for Asthma (GINA) guidelines [11], exacerbations and treatment details including frequency of consultation, rescue therapy, hospitalization, type and dose of asthma medications, respiratory variables relevant to the first visit and their subsequent visits. Data for this study are from the National Research Council of Palermo, a public primary care institution comprising a pneumological clinical. A total of 1930 patients were recorded in the database over the period 2001–2012.

Inclusion criteria were: consecutive lifelong non-smoking asthmatics followed-up over a 10-year period in an outpatient asthma clinic; a follow-up visit every three months for every year of observation, inclusive of pulmonary evaluation with spirometry. We enrolled 100 non-smoking asthmatics (aged 18–66 years, 41 M). The anthropometric, clinical, and functional characteristics at enrolment are presented in Table 1. FEV_1_ rate of decline compared with other studies are presented in Table 2.

At enrolment, personal history of bronchial asthma was confirmed by FEV_1_ reversibility (FEV_1,rev%_) after inhalation of 400 μg salbutamol [11,12]. In the presence of suggestive asthma symptoms without immediate reversibility, subjects underwent a course of oral steroids (OS): a new functional assessment was performed and long-term reversibility computed.

Skin prick tests were performed with a panel including the most common aeroallergens, plus positive and negative control (Stallergenes Italia S.r.l), following EAACI recommendations. Allergic sensitization was indicated by at least one positive skin prick test [13].

During the follow-up, subjects underwent clinical and spirometric evaluations every three months and therapeutic recommendations were repeated. Patients were treated according to GINA guidelines [11]. In case of exacerbation, patients could receive telephonic counselling or unscheduled visit.

An inhaled corticosteroid (ICS) score was computed on the basis of daily ICS dose in µg: Low ≤ 500; Moderate > 500 and ≤1000; High > 1000. Cycles of short-term OS were used for computing the asthma exacerbations. The number of exacerbations during the 1st, 5th, and 10th years of follow-up were recorded. Subjects with a number of OS ≥ 2 were identified as frequent exacerbators, while those with OS < 2 were infrequent exacerbators.

FEV_1_ variability, as expression of bronchial responsiveness [14] was calculated at the 1st, 5th, and 10th year of follow-up, following the formula: (FEV_1_,_max_ − FEV_1_,_min_)/FEV_1_,_pred_ × 100. FEV_1_,_max_ and FEV_1_,_min_ were the maximum and minimum FEV_1_ recorded during the 1st, 5th, and 10th years of follow-up, and FEV_1,pred_ was the corresponding individual predicted FEV_1_.

To compare all subjects independently of height, individual FEV_1_ data were normalized for the subject’s height at the third power (FEV_1_/Ht^3^, L/m^3^/y) [15]. For each year of follow-up, the best FEV_1_ measure in each 6-month period was analysed [16]. Thus, in the follow-up, 20 FEV_1_/Ht^3^ values resulted for each subject. The relationship between FEV_1_/Ht^3^ values as dependent variable and year (or year fractions) as independent variable was treated by linear regression analysis to obtain individual slopes to the 1st (slope FEV_1_/Ht^3^-1st period) and 2nd period (slope FEV_1_/Ht^3^-2nd period). The individual differences between slopes were calculated as differences between the slope of the 6th–10th year period and slope of the 1st–5th year period (∆slope FEV_1_/Ht^3^).

Slopes were tested against the investigated variables (unless otherwise indicated, continuous variables were dichotomized using the median value): gender, age at enrolment (<47.5 and ≥47.5 years), age of disease onset (<29 and ≥29 years), body mass index (BMI, <26 and ≥26 Kg/m^2^), FEV_1_ at enrolment (as continuous variable), FEV_1_ variability (<11.3% and ≥11.3%), reversibility (FEV_1,postBD_ ≥ 12% respect to pre-bronchodilator FEV_1_), disease duration (<13 years and ≥13 years), allergic sensitization (Yes/No), rhinitis (Yes/No), ICS score (Low, Moderate, High) and exacerbation score (frequent/infrequent exacerbators).The study was approved by the Local Institutional Ethics Committee (authorization reference number 7/2013).

### Statistical Analysis

Differences in frequency distribution of variables were evaluated by χ^2^ test, median differences, for not normally distributed variables, were evaluated by U Mann–Whitney tests for unpaired sample and by Wilcoxon test for paired sample.

Correlation between continuous variables was investigated using Spearman Rank Correlation.

All computations were performed by StatView statistical software package (SAS Institute, Cary, NC, USA). A probability level of *p* < 0.05 was selected as statistically significant.

## 3. Results

### 3.1. FEV_1_ Decline

The FEV_1_/Ht^3^ slope values for the two periods were not significantly different for gender (Mann–Whitney *U* test). All FEV_1_/Ht^3^ slopes in the 1st period showed negative values whereas in the 2nd period the slopes were significantly less negative than 1st period slopes or positive (Wilcoxon test, *p* < 0.0001). The median FEV_1_/Ht^3^ slope computed on the whole population sample was −0.010 L/m^3^/year (range −0.079 to −0.0004) for years 1–5 and −0.006 L/m^3^/year (range −0.038 to +0.038) for years 6–10 (Figure 1). In years 1–5, FEV_1_ loss was 42.5 mL/year, computed for a 1.62 m tall subject (median height of sample); it was 25.5 mL/year, in years 6–10. A significant inverse relationship was found between ∆slope FEV_1_/Ht^3^ and slope FEV_1_/Ht^3^ 1–5 year (*p* < 0.0001 Spearman Rank Correlation) (Figure 2).

### 3.2. Determinants of FEV_1_ Decline

A significant inverse correlation was found between the FEV_1_ slopes of years 1–5 and FEV_1_ at enrolment, expressed as percent of predicted (*p* = 0.02, Spearman Rank Correlation); such a correlation was also found for years 6–10 (*p* = 0.01) (Figure 3, Panels A and B).

Subjects with FEV_1_ reversibility showed steeper FEV_1_ slopes in years 1–5 with respect to subjects without reversibility (*p* = 0.01, Mann–Whitney *U* test) but no significant effect of reversibility was observed on FEV_1_ slopes for years 6–10 (Figure 4).

FEV_1_ variability at the 1st year had a significant effect on lung function decline in the 1st period: subjects with FEV_1_ variability ≥12% showed significantly steeper slopes in years 1–5 (*p* < 0.04, Mann–Whitney *U* test) compared to subjects with lower variability. No significant effect on FEV_1_ slopes for years 6–10 was observed for subjects with high and low FEV_1_ variability at 1st year (Figure 5).

A significant reduction in FEV_1_ variability was observed during the follow-up: median FEV_1_ variability at the 5th and 10th years was significantly lower than that of the 1st year (*p* < 0.0001, respectively, Wilcoxon test). In addition, a significant difference was found between the 5th and 10th years (*p* = 0.02, Wilcoxon test). (Figure 6).

Analysing exacerbation score, the prevalence of frequent exacerbators was 12%, 15% and 10%, at 1st, 5th and 10th year, respectively, with a significant difference in frequency distribution between 1st and 10th years (*p* = 0.004, χ^2^), and between 5th and 10th years (*p* = 0.001, χ^2^). No significant difference between 1st and 5th years was found. Frequent exacerbators in the 1st year had steeper FEV_1_/Ht^3^ slopes in the 1st period (*p* = 0.01, Mann–Whitney *U* test) but no effect was found on 2nd period slopes. Analysing ICS scores across the follow-up, we found that the number of subjects using higher doses of ICS was significantly lower at the 10th years respect to the 5th and to the 1st year (*p* < 0.0001, *p* = 0.003; **χ**^2^, respectively) and at the 5th year respect to 1st year (*p* < 0.02; χ^2^), with a complementary increase of subjects using lower doses in the corresponding years. (Figure 7). BMI had no significant effect of age at enrolment. Disease duration, or age of disease onset was observed on FEV_1_ slopes in two periods. Similarly, gender, allergic sensitization and rhinitis showed no effect on FEV_1_ decline.

## 4. Discussion

This longitudinal study was carried out on 100 lifelong non-smoking adult asthmatic outpatients with a well-defined clinical and functional diagnosis of bronchial asthma. They had a clinical and pulmonary function evaluations every 3 months during a 10-year follow-up.

Our results indicate that FEV_1_ decline in treated asthmatics over the follow-up period was not constant, but rather slowed over time. Whereas a steeper decline was observed during the 1st period, the decline was much slower in the 2nd period. Moreover, while FEV_1_ decline in the 1st period was influenced by reversibility, FEV_1_ at enrolment, FEV_1_ variability in the 1st year, and exacerbations in the 1st year, in the second period it was not. Furthermore, we found an overall decrease over the time in the number of subjects using high/moderate daily doses of inhaled steroids.

When decline was separately computed in the two periods, it showed a striking intraindividual difference: mean FEV_1_ loss was 42.5 mL/year in the 1st period and 25.5 mL/year per year in the 2nd. Thus, mean FEV_1_ decline in the 2nd period was 54% lower with respect to the 1st period. Our study produces different results with respect to previous papers. Peat et al. in an 18-year population health survey reported a loss of 50 mL/year in males (average height 1.70 m) with asthma and a loss of 35 mL/year in normal subjects [15]. Moreover, Lange et al., in a 15-year follow-up study [17] found a loss of 38 mL/year in adult asthmatics and 22 mL/year in non-asthmatics. In a 10-year longitudinal study, Burrows et al. (1) found a 70 mL/year mean overall rate of decline in COPD subjects and 65 mL/year in asthmatics. Contoli [18] more recently, in a 5-year follow-up, reported a rate of FEV_1_ decline of 50 mL/year in asthmatics with fixed airflow obstruction and of 18 mL/year in asthmatics with reversible airflow obstruction.

We supposed that important factors could have influenced the differences in the extent of FEV_1_ loss, such as: (I) incorrect diagnosis (mainly asthma vs. COPD) due to limitations of selected methods (e.g., self-reported diagnosis, questionnaire); (II) inclusion of functional values collected during exacerbations; (III) few functional measurements over the follow-up; (IV) analysis performed on the entire follow-up period; (V) variable effect of pharmacological control of bronchoconstriction over time; and (VI) inclusion of current and former smokers.

We tried to overcome these factors by means of (I) a well-defined asthma diagnosis; (II) four spirometries per year, to minimize the “learning” effect and decrease the risk of decline overestimation due to changes in disease control over time; (III) excluding both current and former smokers; (IV) separate analysis of the 1st and 2nd period of the follow-up to highlight the effect of pharmacological control over time.

All patients were lifelong non-smokers in order to avoid the interaction and detrimental effects of smoking [19,20] on the FEV_1_ decline [21]. Moreover, a typical trajectory of age-related FEV_1_ decline were related to a change in the lifestyle related risk factors, BMI and smoking, these have significantly impact aging-related decline of lung function [22].

In agreement with a previous study [23], we found an inverse relationship between FEV_1_ decline in the 1st period and FEV1 at enrolment, the same result was obtained for FEV_1_ decline in the 2nd period: a steeper FEV_1_ decline in subjects with higher baseline FEV_1_ values. We surmised that the decline can no longer be progressive after the FEV_1_ had previously dropped to a considerable extent.

Acute salbutamol reversibility was not associated with FEV_1_ decline. This lack of association could be explained by the presence of airway inflammation. In fact, when considering long term reversibility (evaluated after trial with OS), we found that subjects with reversibility showed a steeper FEV_1_ decline in the 1st period, but not in the 2nd. Ulrik et al. reported similar results observing that a high degree of reversibility was associated with a steeper functional decline in asthmatics over the following 10 years [21]. Vollmer et al. reported that the response to an inhaled bronchodilator correlated with the rate of FEV_1_ decline only in subjects classified as having bronchial hyperresponsiveness [24]. In our study, using FEV_1_ variability at 1st year, as an expression of bronchial hyperresponsiveness, we found that it was the strongest predictor of lung function decline in the 1st period but not in the 2nd, agreeing with previous studies showing that higher airway responsiveness was responsible for accelerated FEV_1_ decline [25,26,27].

We hypothesized that greater variability of lung function over time is a marker of poorly controlled asthma, thus significantly affecting the rate of FEV_1_ decline. We found that the reduction of FEV_1_ decline over time was associated with a progressive reduction in FEV_1_ variability over time. In agreement with Metha et al., we suggest that the reduction in bronchial hyperresponsiveness produced and maintained by a regular ICS treatment may be responsible for the changes [28]

Moreover, a well-defined effect of ICS on the rate of decline in lung function has been reported, while the effect of bronchodilators was less conclusive, so we choose to analyse only the effect of ICS [29,30].

Observational studies, showed a less pronounced decline in FEV_1_ in asthmatics taking ICS compared to those not receiving them [17]. Moreover, an early and regular ICS, introduced when symptoms are mild, was expected to prevent lung function worsening suggesting that ICS could reduce the intensity of airway remodelling and thus produce slower lung function decline [8,31]

Our results demonstrate that the step-down approach to long-term asthma therapy, grants asthma control along with a positive effect on lung function decline. In fact, despite the large increase in the number of subjects assuming a low ICS dose in the 2nd period with respect to 1st one (19 to 38%), the overall FEV_1_ decline resulted less negative [32].

Previous studies have reported a strong effect of ICS to prevent asthma exacerbations [33,34], accordingly, we found a reduction of number of high exacerbators over the follow-up period. We surmised that the significant reduction of FEV_1_ loss in years 6–10 with respect to years 1–5 could be due to early and regular ICS treatment of our patients [29]. We did not find any effect of disease duration on rate of decline in both periods. It is possible that asthmatics may have an excessive functional decline prior to the time of diagnosis, and also in the first years following asthma onset [3]. Accordingly, the studied subjects had a wide range of disease duration, and we could not clearly document which treatment had been adopted.

In the last decades a growing interest was reported in the lung microbiota. Previous studies reported a shift in the lung microbiota during lung diseases, in particular in asthma [35]. Moreover, the lung microbiota is more diverse and abundant in some subjects with asthma [35,36,37,38,39]. It remains a matter of debate whether we should be talking about dysbiosis, stable colonization, or infections of the lungs. Furthermore, the function and causal role of this shift in the lung microbiota in the outcome of asthma remain unclear [40]. Unfortunately, we did not study this aspect in our patients.

Ulrik and Lange showed that the rate of FEV_1_ decline was higher in subjects with recent asthma onset compared to subjects with chronic asthma, as well as in men compared to women [5,6]. Controversies exist relevant to this effect on FEV_1_ decline [41,42] We did not find any influence of gender on FEV_1_ decline in either the 1st or 2nd 5-year period. The gender difference in FEV_1_ loss reported in previous studies could be explained by women’s increased susceptibility to the lung-damaging effects of cigarette smoking and to development of chronic airway obstruction among asthmatics [19,43,44]. Therefore, we have chosen to include only lifelong non-smokers patients. Furthermore, our patients showed a comparable reversibility airflow obstruction that could explain the similar rate of FEV_1_ decline over time.

With regard to the influence of age on lung function decline in asthma, we did not find any influence of age on FEV_1_ decline either in the 1st or 2nd period. Similar results were reported by Peat et al. in a long follow-up study on asthmatics [15]. Conversely, in other studies, aging was found to be associated with a steeper decline in FEV_1_ [8,20].

In our study, BMI did not influence FEV_1_ decline. Conversely, Marcon A et al. [23] had reported a faster FEV_1_ decline in the non-obese compared with the obese: this finding could be due to a lower baseline FEV_1_ and the process of decline may no longer be progressive after FEV_1_ had previously dropped a considerable extent. This different result could be due to presence of more obese subjects in their sample (BMI > 30).

In agreement with previous studies [15,41] we found that allergic sensitization does not appear to be a determinant of changes in the rate of functional decline in asthma, suggesting that inflammatory processes in the airways of patients with asthma may run their course, irrespective of allergic status.

## 5. Conclusions

The present study shows that the rate of FEV_1_ decline over ten years follow-up in non-smoking adults treated for asthma proves not to be constant when calculated separately in 2 consecutive 5-year periods. In fact, it reduces over time, slowing down until it reaches an FEV_1_ rate of decline comparable to normal subjects. In the 1st 5-year period, FEV_1_ reversibility, higher FEV_1_ variability and exacerbations are determinant factors, while in the 2nd period those factors are no longer determinant.

These findings suggest the possible role of early, continuous, and regular long-term treatment with ICS in reducing number of high exacerbators over the follow-up period, the intensity of airway remodelling and thus produce slower lung function decline in asthma patients [45].

## Figures and Tables

**Figure 1 diagnostics-11-01637-f001:**
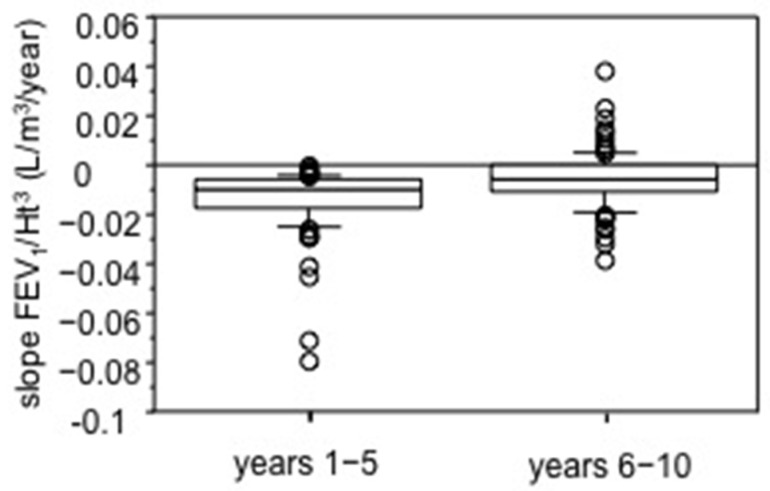
Slope values of relationships between height-adjusted FEV_1_ and time (L/m^3^/year), separately for years 1–5 and 6–10. Bars indicate (from the bottom to the top) 10th, 25th, 50th (median), 75th and 90th percentiles. Values below 10th and above 90th percentiles are plotted as circles. A significant difference was found (*p* < 0.0001, Wilcoxon test).

**Figure 2 diagnostics-11-01637-f002:**
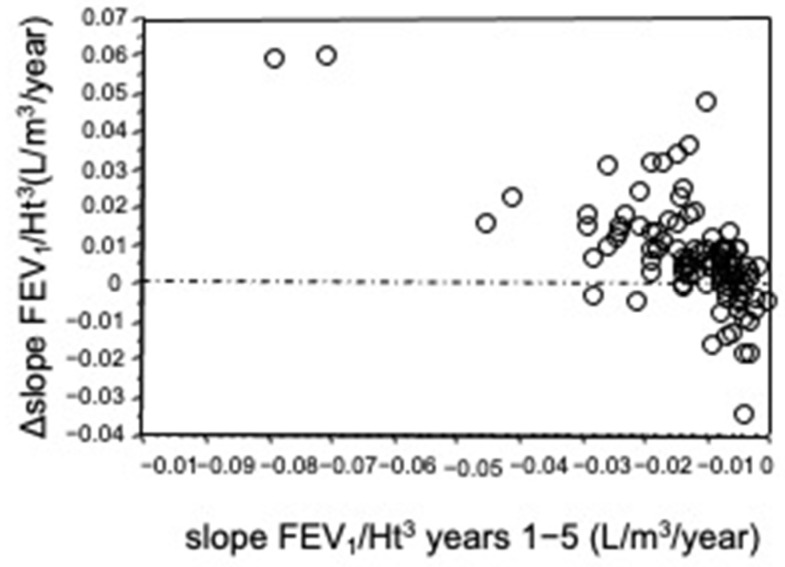
Relationship between the individual differences between the slope of the 6th–10th year period and slope of the 1st–5th year period (∆slope FEV_1_/Ht^3^) and slope FEV_1_ years 1–5. A significant inverse correlation was found (*p* < 0.0001, Spearman Rank Correlation).

**Figure 3 diagnostics-11-01637-f003:**
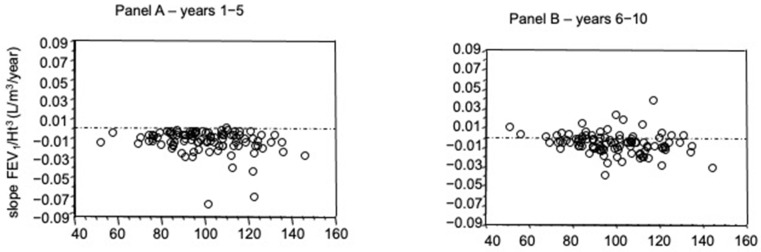
Relationships between slope values of relationships between height-adjusted FEV_1_ and time (L/m^3^/year) for years 1–5 (**Panel A**) and for years 6–10 (**Panel B**) and FEV_1_ at enrolment (% of predicted). Both correlations were significant (*p* = 0.02 and *p* = 0.01, respectively, Spearman Rank Correlation).

**Figure 4 diagnostics-11-01637-f004:**
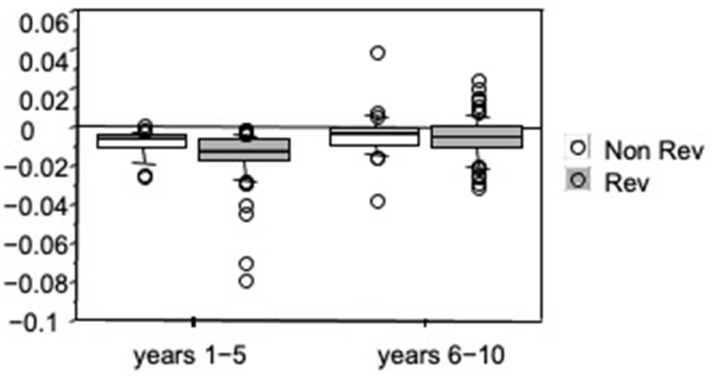
Slope values of relationships between height-adjusted FEV_1_ and time (L/m^3^/year) during years 1–5 and years 6–10, separately for subjects with or without FEV_1_ reversibility at enrolment. Bars indicate (from the bottom to the top) 10th, 25th, 50th (median), 75th and 90th percentiles. Values below 10th and above 90th percentiles are plotted as circles. In years 1–5 subjects with FEV_1_ reversibility at enrolment showed significantly lower slope values (*p* = 0.01, Mann–Whitney *U*-test). No significant effect of reversibility was observed on FEV_1_ slopes for years 6–10.

**Figure 5 diagnostics-11-01637-f005:**
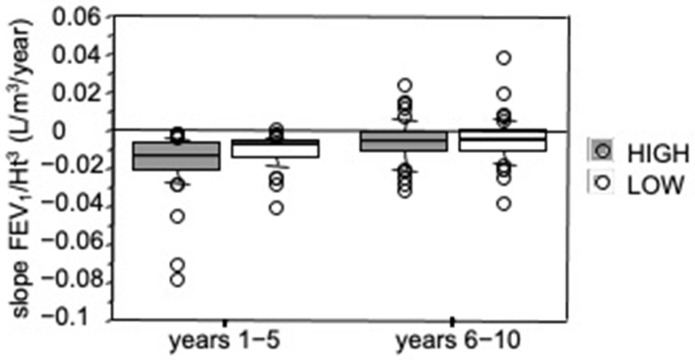
Slope values of relationships between height-adjusted FEV_1_ and time (L/m^3^/year) during years 1–5 and years 6–10, separately for groups of FEV_1_ variability at 1st year. Bars indicate (from the bottom to the top) 10th, 25th, 50th (median), 75th and 90th percentiles. Values below 10th and above 90th percentiles are plotted as circles. In years 1–5 subjects with FEV_1_ variability at 1st year showed significantly lower slope values (*p* < 0.04, Mann–Whitney *U*-test). No significant effect of variability was observed on FEV_1_ slopes for years 6–10.

**Figure 6 diagnostics-11-01637-f006:**
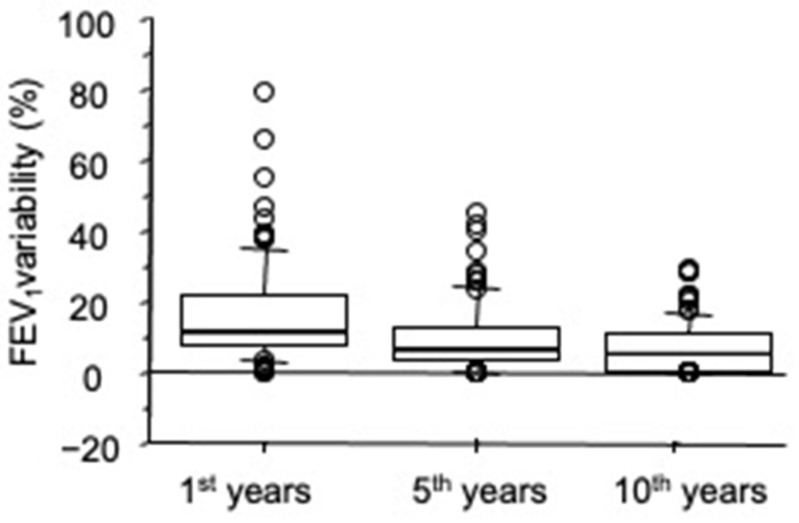
FEV_1_ variability (in percent of predicted), separately for 1st, 5th, and 10th years. Bars indicate (from the bottom to the top) 10th, 25th, 50th (median), 75th, and 90th percentiles. Values below 10th and above 90th percentiles are plotted as circles. Significant differences were found between 1st and 5th years, between 1st and 10th years and between 5th and 10th years (Wilcoxon test).

**Figure 7 diagnostics-11-01637-f007:**
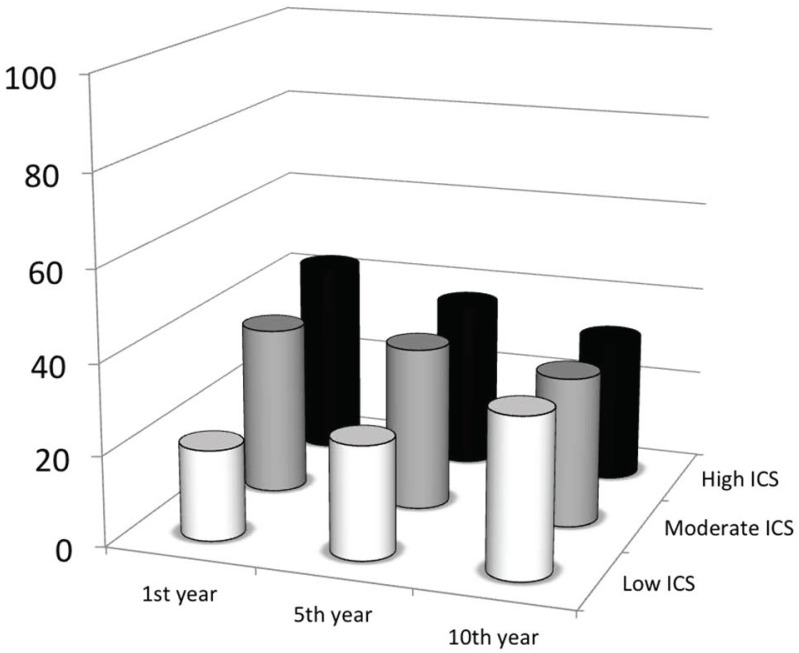
Prevalence of subjects, for each inhaled corticosteroid (ICS) score during the follow-up period. A Scheme of 10th and 5th years (*p* < 0.0001, χ^2^), between 10th and 1st years (*p* = 0.003, χ^2^), and between 5th and 1st years (*p* < 0.02, χ^2^).

**Table 1 diagnostics-11-01637-t001:** Anthropometric, clinical, and pulmonary functional characteristics at enrolment, according to sex.

	Females (No 59)	Males (No 41)	*p* Value
Age at enrolment	49 (35–55)	43 (30–53)	0.04
Body mass index	25 (22–30)	27 (23–28)	N.S.
Age of disease onset	33 (19–43)	24 (12–38)	N.S.
Disease duration	12 (7–21)	15 (5–20)	N.S.
FEV_1_ at enrolment	85 (70–97)	80 (70–91)	N.S.
FVC at enrolment	105 (93–114)	98 (96–109)	N.S.
FEV_1_/FVC at enrolment (absolute, %)	69 (61–75)	68 (58–73)	N.S.
FEV_1_ rev, %	16 (9–33)	23 (12–51)	N.S.
Allergic sensitization (No, %)	41 (69)	28 (68)	N.S.
Rhinitis (No, %)	40 (68)	26 (63)	N.S.
Stage 1 Gina treatment (No, %)	12 (20)	8 (19)	N.S.
Stage 2 Gina treatment (No, %)	22 (37)	15 (37)	N.S.
Stage 3 Gina treatment (No, %)	25 (43)	18 (44)	N.S.

Data are presented as median and IQ range unless noted otherwise, separately for gender. None of the differences was significant between male (M) and female (F) subgroups, (Mann–Whitney *U*-test, χ^2^ test) except for Age at enrolment.

**Table 2 diagnostics-11-01637-t002:** Overall FEV_1_ rate of decline compared with other studies. FEV_1_ decline values were separately calculated for a male and a female.

	Reference		Asthma	COPD
	M	F		
Quanjer (Bull Eur Physiopath Respir 1983)	29 mL/year	25 mL/year		
Lange N Engl J Med 1998			38 mL/year	
Peat Eur J Respir Dis 1987			50.5 mL/year	
Fletcher 1976 (libro)			22 mL/year	
Cuttitta Chest 2002			40	41.3 mL/year
Burrows 1986			65 mL/year	70 mL/year
Mannino Soriano (Am J Respir Crit Care Med 2009)	18 mL/year			
Kalhan R (Am J Med 2010) Framingham	19.6	17.6 mL/year		
O’ Byrne PM (Am J Respir Crit care Med 2009)	27–34 mL/year			

## Data Availability

Primary data are available upon request.

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
