# Peer review of "Lung Function Decline in Adult Asthmatics—A 10-Year Follow-Up Retrospective and Prospective Study"

_diagnostics, 2021, doi:10.3390/diagnostics11091637_

Round 1

Reviewer 1 Report

The authors aimed I-) to describe the changes in forced expiratory volume in one second (FEV1) in lifelong nonsmoking adult asthmatic outpatients during a 10-year follow-up comparing years 1-5 (1st period) with years 6-10 (2nd period); II) to assess factors affecting these changes.

The study covers some issues that have been overlooked in other similar topics. The structure of the manuscript appears adequate and well divided in the sub-paragraphs. Moreover, the study is easy to follow, but some issues should be improved.

The manuscript needs moderate grammar correction. Please also check typos thorough the text.

Conclusion Section: This paragraph is included in the discussion section. Please separate same. Moreover, is also required a general revision to eliminate redundant sentences and to add some "take-home" message.

Author Response

Dear Reviewer,

Thank you for your attention to this matter.
In the revision a check typos throughout text was now performed. Conclusion session and discussion were adequately separated. Moreover, we eliminated and added some "take-home" message.

Sincerely

Reviewer 2 Report

The retrospective and prospective study on the DECLINATION OF LUNG FUNCTION in ADULT ASTHMA. It's very interesting.
My comments.
it seems strange to me that there are no sex differences, the authors can explain better.
Do the authors think the microbiota can affect? are there any studies?
I suggest a graphical abstract, so that the results are more immediate

Author Response

Dear Reviewer,
Thank you for your attention to this matter.

In revision tex, we
added a comment to explain our results concerning question about gender differences. We  discussed the microbiote role in discussione in asthma and abstract was modified with a grafical style.

Sincerly